# Nasogastric Tube Feeding in Anorexia Nervosa: A Propensity Score-Matched Analysis on Clinical Efficacy and Treatment Satisfaction

**DOI:** 10.3390/nu16111664

**Published:** 2024-05-29

**Authors:** Matteo Martini, Paola Longo, Clara Di Benedetto, Nadia Delsedime, Matteo Panero, Giovanni Abbate-Daga, Federica Toppino

**Affiliations:** Eating Disorders Center, Department of Neuroscience “Rita Levi Montalcini”, University of Turin, Via Cherasco 11, 10126 Turin, Italy; matteo.martini@unito.it (M.M.); paola.longo@unito.it (P.L.); clara.dibenedetto@unito.it (C.D.B.); nadia.delsedime@unito.it (N.D.); matteo.panero@unito.it (M.P.); federica.toppino@unito.it (F.T.)

**Keywords:** enteral feeding, nasogastric tube, nutritional rehabilitation, anorexia nervosa, eating disorders, treatment satisfaction

## Abstract

The choice of a refeeding strategy is essential in the inpatient treatment of Anorexia Nervosa (AN). Oral nutrition is usually the first choice, but enteral nutrition through the use of a Nasogastric Tube (NGT) often becomes necessary in hospitalized patients. The literature provides mixed results on the efficacy of this method in weight gain, and there is a scarcity of studies researching its psychological correlates. This study aims to analyze the effectiveness of oral versus enteral refeeding strategies in inpatients with AN, focusing on Body Mass Index (BMI) increase and treatment satisfaction, alongside assessing personality traits. We analyzed data from 241 inpatients, comparing a group of treated vs. non-treated individuals, balancing confounding factors using propensity score matching, and applied regression analysis to matched groups. The findings indicate that enteral therapy significantly enhances BMI without impacting treatment satisfaction, accounting for the therapeutic alliance. Personality traits showed no significant differences between patients undergoing oral or enteral refeeding. The study highlights the clinical efficacy of enteral feeding in weight gain, supporting its use in severe AN cases when oral refeeding is inadequate without adversely affecting patient satisfaction or being influenced by personality traits.

## 1. Introduction

Anorexia Nervosa (AN) is a complex mental disorder characterized by severe dietary restrictions, an intense fear of gaining weight, and a distorted body image. Its origins are multifaceted, involving psychological, physiological, and sociocultural factors [1,2]. The Diagnostic and Statistical Manual of Mental Disorders, Fifth Edition (DSM-5), identifies two subtypes of AN: the restricting type (AN-R), where weight loss is achieved through dieting, fasting, and/or excessive exercise, and the binge–purging type (AN-BP), which includes episodes of binge eating and/or purging behavior, such as self-induced vomiting or the misuse of laxatives or diuretics [3]. AN profoundly disrupts cognitive, emotional, and social functioning and leads to severe medical complications across multiple systems, including cardiovascular, gastrointestinal, endocrine, and cerebral [4,5]. It also frequently presents comorbidity with mood, anxiety, and personality disorders [6,7].

AN poses challenges in psychiatric as well as in nutritional management, particularly in the choice of refeeding strategies, which significantly impact patient recovery [8]. Effective nutritional management is pivotal in hospital-based treatments aimed at restoring nutritional health and addressing psychopathological aspects of the disorder [9,10]. Gaining weight during hospitalization is crucial for recovering from AN and is linked to favorable outcomes after discharge [11,12]. Traditionally, oral refeeding and enteral refeeding through a Nasogastric Tube (NGT) have been employed as critical interventions in managing severe cases of eating disorders, both in adult and pediatric settings [13,14]. Enteral nutrition can be employed both in the case of AN-R and AN-BP [15] but also occasionally in other eating disorders (EDs) with restrictive patterns, such as Atypical Anorexia [16] or Avoidant Restrictive Food Intake Disorder (ARFID) [17], or more rarely in Bulimia Nervosa (BN) to reduce purging behaviors [18].

The literature typically recommends oral, enteral nutrition as the first line of treatment, resorting to NGT only when purely oral strategies fail [8,19]. Whereas oral refeeding allows patients to regain control over their eating in a supportive environment, NGT refeeding is often reserved for individuals who are at significant medical risk or who do not respond to oral refeeding plans [20]. According to the most recent American Psychiatric Association (APA) guidelines, NGT feeding can be employed for acute nutritional rehabilitation in patients who fail to meet their prescribed caloric intake through meals alone and could also be influenced by factors such as age, other clinical characteristics, and the availability of specialized treatment programs like meal-based behavioral treatment [8]. Medical Emergencies in Eating Disorders (MEED) guidelines agree with using NGT only as a second resort and specify that it can become necessary when a patient cannot consume enough nutrients orally or cannot eat by mouth [21]. However, there are subtle differences in the approach to severely ill patients: the APA guidelines suggest that the decision to use NGT should not be solely based on medical instability or the severity of illness, whereas the MEED guidelines specifically list clinical/biochemical instability and life-threatening weight loss (Body Mass Index or BMI < 13) as criteria for NGT use. Enteral feeding, hence, sometimes becomes associated with nutrition under restraint [22], a rare but necessary approach in case of rapid health decline and severe psychopathological impairments where patients fail to recognize their illness’s severity or engage in treatment. Moreover, NGT has been recognized for ensuring rapid nutritional restoration and mitigating immediate health risks like refeeding syndrome (RS), a potentially fatal medical complication [19]. However, the literature presents a wide range of refeeding regimes and reasons for initiation, alongside mixed results on the efficacy of enteral feeding versus purely oral nutrition in achieving weight gain [23,24], with long-term effects remaining ambiguous [10,25].

Furthermore, the invasive nature of enteral feeding has been speculated to aggravate body image psychopathology and potentially weaken the essential patient–psychiatrist therapeutic alliance [26,27]. Also, some studies have evidenced the risk of non-adherence, with nearly 30% of patients manipulating the refeeding tube [15], while others have evidenced that nasogastric feeding is well tolerated [28,29]. Among the limited studies that have examined the psychological aspects of NGT, there is the one from Zuercher and colleagues [30]: this retrospective study not only assessed the medical complications and effects on weight gain associated with voluntary enteral nutrition in patients with AN but also explored patient satisfaction with treatment, finding no significant differences in psychological outcomes between patients who were tube-fed and those who were orally fed. However, apart from this study, the literature remains sparse on topics such as treatment satisfaction and the impact of enteral feeding on the therapeutic alliance.

Enteral feeding in the treatment of AN could also potentially intersect with aspects of personality traits that are prevalent among individuals with the disorder. Studies have indicated that traits such as perfectionism, rigidity, and a high need for control are commonly observed in individuals with AN [31,32,33]. These personality characteristics can influence how patients perceive and respond to treatment modalities like NGT, which may feel invasive or coercive. However, to our knowledge, the literature has yet to explore the relationship between personality traits and enteral feeding. It is also worth noting that a good part of studies comparing enteral to oral feeding are conducted by clinicians specialized in nutrition or dieticians [26,34] rather than psychiatrists or clinical psychologists, highlighting a gap in data regarding the emotional impacts of enteral nutrition.

In this study, we aim to compare a group of AN individuals who underwent NGT nutritional treatment during hospitalization with a group of propensity scores that matched AN individuals who proceeded to entirely oral nutritional rehabilitation. In this way, we aim to analyze with a quasi-experimental method the effect of enteral therapy on (1) BMI increase and (2) treatment satisfaction, accounting for therapeutic alliance. The secondary aim is to assess baseline personality differences (especially affective temperaments and perfectionism) in the two groups.

We adopted propensity score matching because this method offers a way to retrospectively balance baseline characteristics of individuals who were not exposed to treatment to those who received the treatment, thus mimicking the randomization process and allowing for more accurate comparisons. In fact, randomized controlled trials are not always feasible for ethical reasons, and in many cases, the generalizability of RCTs to real-world scenarios is limited. Conversely, conclusions that can be drawn from purely observational studies are often hindered by confounding factors acting both at the exposure and outcome levels.

Based on the previous literature, we expected NGT treatment to be associated with greater BMI improvement during the hospital stay in comparison to matched individuals who did not receive enteral feeding. Predictions regarding the subjective outcomes of NGT are more difficult due to the scarcity of data; however, we expected no significant differences in treatment satisfaction in the two groups.

## 2. Materials and Methods

### 2.1. Participants

This observational study is based on data collected from patients hospitalized in the Eating Disorder Center of “Città della Salute e della Scienza” hospital at the University of Turin from February 2014 to February 2024. The inclusion criteria were as follows: (a) female gender; (b) diagnosis of AN assessed by an experienced psychiatrist through the Structured Clinical Interview for Diagnostic and Statistical Manual of Mental Disorders, 5th edition (DSM-5) [35]; (c) age > 18 years old and <65 years; (d) no current psychotic, bipolar, or active substance use disorders; (e) written consent for ethical approval of the study. All patients were assessed at admission, with weight and height measurements taken by a trained nurse; weight was also measured at discharge. Psychiatric comorbidity (i.e., most commonly the presence of anxious, depressive, obsessive-compulsive disorder, or a diagnosis of personality disorder) was evaluated at admission by the psychiatrist conducting the clinical interview.

Patients completed questionnaires within five days following admission, while end-of-treatment questionnaires covering eating symptomatology and treatment alliance were administered before discharge. Questionnaires assessing personality features were generally administered only during the first hospitalization. Only individuals with a complete eating symptomatology questionnaire completed at baseline were included in this study.

Given the high incidence of frequent relapses and multiple hospitalizations associated with AN [36], to ensure a sufficiently large sample, up to three hospitalizations per individual were included, with this parameter accounted for in the analyses. Then, a total of 340 hospitalizations for 241 inpatients with AN were considered in this study. After the matching balance for confounding factors as described below, we included 194 hospitalizations for 134 individuals.

The study was conducted according to the principles of the Declaration of Helsinki and approved by the hospital Ethical Committee (approval number 0036472).

### 2.2. Intervention

Admissions to the inpatient ward typically followed emergency department referrals or were decided when outpatient treatments had proven insufficient. A multidisciplinary team composed of psychiatrists, nurses, dieticians, and a general clinician provided continuous care throughout the inpatient stay. According to international guidelines [8], oral nutritional rehabilitation was the preferred refeeding method; however, enteral therapy was considered for severe cases, typically when significant nutritional intake challenges were encountered early in hospitalization. The decision to prescribe nutritional therapy via NGT was discussed within the multidisciplinary team and with the patient, and NGT therapy was administered by clinicians with years of expertise in the inpatient treatment of AN. In this case, NGT with daytime continuous nutritional supplementation was implemented along with the usual five oral meals per day in an integrated manner. About 95% of patients with NGT commenced both oral and enteral nutrition at the start of their hospitalization, whereas the remaining 5% initially received only enteral feeding and transitioned to combined rehabilitation after a few days. The caloric intake at admission, both in the case of oral or enteral rehabilitation, was decided individually by the general clinician and the dietician based on caloric intake at home, weight, and nutritional parameters. Psychopharmacological therapy, including the use of antidepressants, antipsychotics, and anxiolytics, was frequently used for comorbid symptoms such as depression and anxiety. Discharge criteria were based on clinical stability, with patients either transitioning to specialized clinics for ongoing weight restoration treatment or returning home for outpatient follow-up.

### 2.3. Measures

The following instruments were used for assessment:Eating Disorders Examination Questionnaire (EDE-Q) [37], evaluating the frequency of typical ED behaviors in the last 28 days and the severity of various aspects of ED psychopathology. It is composed of four subscales (restraint, eating concern, shape concern, and weight concern) and a global score. Internal consistency (Cronbach alpha) was 0.96.Frost Multidimensional Perfectionism Scale (FMPS) [38,39], a self-report questionnaire consisting of 35 questions designed to assess perfectionism across a total score plus four sub-scales: concern over mistakes and doubts about actions; excessive concern with parents expectations and evaluation; excessively high personal standards; concern with precision, order, and organization. Internal consistency was 0.94.Temperament Evaluation of Memphis, Pisa, Paris, and San Diego Autoquestionnaire (TEMPS-A) [40], a clinical tool for evaluating temperament that consists of 110 items. Internal consistency (Cronbach alpha) was 0.91.Working Alliance Inventory-Short Revised (WAI-SR) [41], a test designed to address the therapeutic alliance through three dimensions: (a) agreement on the tasks of therapy (task), (b) agreement on the goals of therapy (goal), and (c) development of an affective bond between patient and therapist (bond). Internal consistency (Cronbach alpha) was 0.92.Treatment Evaluation and Acceptability Questionnaire: various aspects of the treatment, like usefulness, satisfaction, subjective improvement, meal assistance, and appropriateness of length of stay, were evaluated by administering Visual Analog Scales (VAS) from 0 to 10. Internal consistency (Cronbach alpha) was 0.77.

### 2.4. Data Analysis

#### 2.4.1. Missing Data

Analyses were run in R version 4.3.0 [42] using RStudio [43]. As per the study design, there were no missing data in the admission eating psychopathology questionnaire. Regarding personality variables, 7 individuals in the NGT group and 10 in the non-NGT group had missing questionnaires. Regarding discharge variables, after the matching described below, 27 hospitalizations in the NGT and 28 in the non-NGT group had end-of-treatment questionnaires missing. Comparing completers versus non-completers, no significant differences emerged in baseline and end-of-treatment variables (Appendix A) between the NGT and non-NGT group, except for higher values of EDE-Q weight concern in the NGT group (*p* = 0.042; Appendix A).

#### 2.4.2. Causal Assumptions

As recommended in the recent literature [44], we specified our assumptions in a Directed Acyclic Graph (DAG) using the web interface of the package dagitty [45].

DAGs are visual representations of the variables that researchers assume to be responsible for the data-generation process of interest [44]. In such graphs, variables are represented as nodes, and the causal relationships between them are arrows. Such representation allows for defining the causal path that moves from the exposure to the outcome, as well as the so-called biasing paths due to confounders. Such paths could arise, for instance, when both the exposure and the outcome are caused by a third variable. These relationships need to be taken into account in the analysis in order to avoid biased estimates of the effect of exposure on the outcome.

In simple terms, DAGs provide a way to (1) transparently state assumed causal relations and (2) identify confounders in the causal relationship of interest. As a practical implication, by inspecting a DAG (or using programs dedicated to drawing and analyze DAGs such as dagitty) [44], researchers can easily define the sufficient adjustment set for their analysis (i.e., the variables to condition on in order to “close non-causal paths”).

The assumptions that guided our study and the selection of control variables for the regression models are therefore contained in the DAG represented in Appendix A.

In this study, we were interested in modeling the direct effect of enteral therapy on the considered outcomes. As can be appreciated in the DAG, a modeling of the total effect of enteral therapy on BMI increase is not feasible unless the proximal reason for Nasogastric Tube positioning is taken into account (i.e., difficulty in nutrition per os during the first days of treatment). Conversely, the direct effect can be assessed when the other confounders are taken into account.

Regarding the variables influencing treatment evaluation, our assumptions allow us to model the effect of enteral therapy on this outcome while accounting for therapeutic alliance and personality features. Conversely, the evaluation of the effect of enteral therapy on therapeutic alliance would be biased since we only measured this variable at the end of treatment and could not account for individual differences in early working alliance.

#### 2.4.3. Propensity Score Matching

In order to investigate the effect of enteral therapy on the outcomes in balanced groups, we calculated the propensity score using the following baseline variables: age; duration of illness; BMI; AN subtype; anxious, depressive, and OCD comorbidity; personality disorder comorbidity; caloric intake; EDE-Q total score; and number of hospitalization. These variables were selected according to evidence from the literature and clinical expertise [46,47,48].

Propensity score calculation and matching were performed with the package MatchIt [49], using the method “nearest” and 1:1 match with a threshold for the standard mean differences of 0.1.

#### 2.4.4. Group Comparisons and Regression Models

We compared the groups before and after matching on baseline and end-of-treatment variables.

Regression models were conducted in the matched groups with BMI increase and treatment subjective evaluation as dependent variables.

The control variables were selected according to the sufficient adjustment set that emerged from the DAG (see Appendix A). For BMI increase, these were baseline BMI, caloric intake at discharge, reduction in EDE-Q total score, and length of stay. For treatment satisfaction, the variables to control for were BMI increase, reduction in EDE-Q total score, length of stay, personality features, and therapeutic alliance measured at discharge. Personality variables selected were perfectionistic concern over mistake and depressive temperament based on the previous literature [31,50,51].

Regarding treatment evaluation, we chose to analyze the reports from the individuals regarding (1) usefulness, (2) satisfaction, and (3) subjective improvement.

## 3. Results

Balancing was successful for all the variables considered (see Appendix A). As can be seen in Table 1, before matching, individuals who were prescribed enteral therapy presented at admission with more severe caloric restrictions and more severe eating psychopathology as measured by EDE-Q. However, these differences were no longer present after matching.

Similarly, at discharge (Table 2), before matching, the length of stay was longer for the NGT group but not significantly different after matching. Conversely, even after matching, the mean BMI increase remained significant, with a higher increase for the NGT group. No other significant differences were present between the two groups at discharge, with the exception of a higher level in the NGT group in the subscale regarding the bond between the patient and the clinician, both in the unmatched and matched samples. The mean duration of NGT treatment during hospitalization was 25 days.

The regression models built following the adjustment set described in the causal model took into account the length of stay and the other potential confounders of the relationship between NGT and BMI increase (Table 3) and subjective treatment usefulness, satisfaction, and meal assistance (Table 4).

From these models, enteral therapy has a significant effect on BMI increase, whereas no significant effects emerge on individuals’ evaluations of the treatment. Working alliance at discharge was significantly and positively associated with all three EOT questionnaire outcomes. R2 of the first model (BMI increase) was 0.36, whereas for the subsequent models, it was 0.38, 0.42, and 0.39. Model diagnostic showed no issues in the regression models (Appendix A).

Finally, a comparison of personality features in the treated vs. non-treated group showed no significant differences (Appendix A).

## 4. Discussion

The present study aimed to evaluate the effects of enteral therapy on BMI increase and treatment satisfaction in inpatients with AN. Through propensity score matching, we ensured balanced comparison groups for key variables (age, illness duration, diagnostic subtype, BMI, psychiatric comorbidity, caloric intake, total EDE-Q, and number of hospitalizations). 

Our regression analysis revealed a significant impact of enteral therapy on BMI increase, highlighting efficacy in nutritional rehabilitation. Specifically, the presence of enteral therapy positively correlated with BMI increase, underscoring its potential to facilitate more substantial weight recovery in patients with AN. Although a minority of studies [25] disagree on the effect of enteral therapy on weight, our finding aligns with the majority of previous research [9,15,23], which evidences that NGT enables reaching a generally higher caloric intake [26]. However, as much as enteral feeding seems effective on short-term weight gain, the literature on the effect on long-term ED symptomatology [15,25] and long-term weight outcomes [26] is still scarce. 

Our regression analyses also show no differences between the two groups regarding treatment satisfaction accounting for the therapeutic alliance. The majority of the literature agrees on the positive effect of enteral therapy on physical health parameters, like ensuring a relatively rapid weight restoration while managing the risk of RS [9]. However, its influence on psychological outcomes, including treatment satisfaction, has not yet been thoroughly ascertained. Culturally, tube feeding has been associated with coercion in treatment, with recent studies deepening this aspect [34,52], creating debate around the ethics of its usage [53,54]. Evidence on the psychological effects of NGT is still scarce [15], but some studies are starting to assess the tolerability and acceptability of this kind of treatment [26]. A recent qualitative study [28] evidenced the importance of involving the patient in the decision about enteral treatment, communication, and addressing phobic attitude towards solid food: most of the patients described themselves as “not bothered” by the tube. Also, the attitude towards NGT from the staff was researched, highlighting some ambivalence between adherence to established guidelines and ethical concerns. Until now, however, only one study [30], to our knowledge, has researched treatment satisfaction after feeding with NGT, with no replication after. This study, like ours, compared inpatients with and without NGT and evidenced no differences in satisfaction with treatment measured with a questionnaire compiled at discharge.

Notable findings are the significant bond between the patient and the clinician in the NGT group at discharge and the strong and positive link between the Working Alliance Inventory (WAI) total score and how patients view their treatment across our model. This shows that the relationship between patient and clinician plays a crucial role in how satisfied patients feel with their treatment, regardless of the specific treatment type. This emphasizes the importance of a good therapeutic relationship in treating AN, as assessed in the previous literature [55,56,57], characterized by transparent communication, involving patients in decision-making processes, and providing ongoing emotional and psychological support to address concerns related to body image and eating control. However, these findings are initial, and future research should examine the therapeutic relationship from the start of treatment to fully understand its impact on forms of treatment like enteral nutrition.

Our findings also suggest that there are no differences in assets of personality characteristics, measured at admission, between patients who undergo nutrition with NGT and those who maintain only oral refeeding practice. This measure stems from the idea that often, in clinical practice, the decision for enteral nutrition derives not only from medical parameters but also from the attitude of the patients towards oral intake and treatment in general, potentially hiding different personality traits. Scientific research has already assessed the link between perfectionism, obsessive traits, negative emotionality, harm avoidance, anxious temperament, alexithymia, and restrictive EDs, while the binge purging subtype has been prevalently linked to emotional dysregulation [31,32,33,58]. This often leads to different comorbidity, with subjects with AN-R suffering more of a comorbid Avoidant Personality Disorder and subjects with AN-BP frequently diagnosed with Borderline Personality Disorder [59]. This made us wonder if a different pattern of temperament and perfectionism, as registered at the TEMPS and FMPS, could lead to a different adherence to the enteral tube, but this was not confirmed in our analyses. This could mean that the reasons for implementing enteral nutrition depend prevalently on clinical parameters and subjective reasons relating to difficulties in engaging with an entirely solid refeeding program from both the perspectives of the clinician and the patient. However, it is possible that other personality facets or questionnaires (such as the Temperament Character Inventory) should be adopted to grasp the differences between the NGT group and the non-NGT group. Our initial results contribute to and stimulate the ongoing debate about the role of personality traits in the treatment of eating disorders [60,61]. Further deepening and integrating the understanding of these personality traits into the treatment plan could help tailor interventions that not only address the physical health needs but also respect and work with the psychological asset of the patient [62].

The limitations of the present study include its observational design, even though we controlled for many potential confounders through propensity score matching and subsequent regression analyses. Additionally, missing data at discharge could weaken our conclusions, as it is unclear if non-compliance with final evaluations might reflect issues like poorer therapeutic alliance, reduced treatment satisfaction, or personality differences. However, our comparison of completers versus non-completers showed no significant differences in key variables. Furthermore, the reliance on self-reported measures for psychological outcomes may introduce bias, but the use of standardized self-reported measures, validated for assessing psychological outcomes, ensures a degree of reliability and consistency. The decision on NGT positioning was made by clinicians and did not follow standardized decisional algorithms; however, we focused our analysis on the effect of enteral therapy on the outcomes while taking into account potential baseline and end-of-treatment confounders. Regarding caloric intake composition, we do not provide information on the percentage of calories introduced, respectively, via enteral therapy and per os in the NGT group. Finally, only female individuals were included in the study, thus limiting the generalizability of the findings to the male population.

The strength of the study is that it is a more comprehensive evaluation of enteral therapy, including not only physical health parameters but also psychological features, such as treatment satisfaction, therapeutic alliance, temperament traits, and perfectionism. Furthermore, the application of propensity score matching to ensure balanced comparison groups for key variables is a methodological strength. 

Future research should include post-discharge longitudinal studies to assess the long-term effects of enteral therapy on weight maintenance, eating disorder symptomatology, and psychological well-being. Moreover, research focusing on the development of the therapeutic alliance early in the treatment process could shed light on its influence on treatment outcomes. 

## 5. Conclusions

In conclusion, while our study confirms the efficacy of enteral therapy in promoting BMI increase as a crucial aspect of AN treatment, it also sheds light on the complex interplay between clinical outcomes, psychological improvements, and the therapeutic relationship in determining the individuals’ evaluation of treatment. The lack of significant differences regarding treatment satisfaction suggests that enteral therapy is a valid instrument for the integrated treatment of AN inpatients.

## Figures and Tables

**Table 1 nutrients-16-01664-t001:** Admission variables.

	Total Sample			Matched		
Characteristic	NGT, *n* = 97 ^1^	Non-NGT, *n* = 243 ^1^	*p*-Value ^2^	NGT, *n* = 97 ^1^	Non-NGT, *n* = 97 ^1^	*p*-Value ^2^
diagnosis			0.8			0.5
AN-BP	33 (34%)	79 (33%)		33 (34%)	37 (38%)	
AN-R	64 (66%)	164 (67%)		64 (66%)	60 (62%)	
admission BMI	13.99 (1.76)	14.37 (1.91)	0.085	13.99 (1.76)	14.01 (1.89)	>0.9
weight (kg)	37.0 (6.0)	37.7 (5.9)	0.3	37.0 (6.0)	37.0 (5.5)	>0.9
admission caloric intake	600 (378)	798 (382)	<0.001	600 (378)	633 (335)	0.5
number of hospitalization			0.2			0.3
1	65 (67%)	176 (72%)		65 (67%)	69 (71%)	
2	27 (28%)	47 (19%)		27 (28%)	19 (20%)	
3	5 (5.2%)	20 (8.2%)		5 (5.2%)	9 (9.3%)	
duration of illness (years)	7 (9)	7 (8)	0.6	7 (9)	7 (8)	0.8
psychiatric comorbidity	43 (44%)	93 (38%)	0.3	43 (44%)	45 (46%)	0.8
personality disorder	22 (23%)	43 (18%)	0.3	22 (23%)	18 (19%)	0.5
EDE-Q restraint	3.72 (2.10)	3.31 (2.01)	0.10	3.72 (2.10)	3.88 (1.94)	0.6
EDE-Q eating concern	3.39 (1.56)	2.89 (1.58)	0.009	3.39 (1.56)	3.37 (1.52)	>0.9
EDE-Q shape concern	4.62 (1.64)	4.01 (1.70)	0.002	4.62 (1.64)	4.62 (1.53)	>0.9
EDE-Q weight concern	4.20 (1.78)	3.48 (1.84)	0.001	4.20 (1.78)	4.13 (1.68)	0.8
EDE-Q global score	3.98 (1.65)	3.42 (1.65)	0.005	3.98 (1.65)	4.00 (1.53)	>0.9

^1^ *n* (%); mean (SD); ^2^ Pearson’s chi-squared test; Welch two-sample *t*-test; abbreviations: AN-R = Anorexia Nervosa-restricting type; AN-BP = Anorexia Nervosa binge–purging type; BMI = Body Mass Index; EDE-Q = Eating Disorder Examination Questionnaire; NGT = Nasogastric Tube.

**Table 2 nutrients-16-01664-t002:** Discharge variables.

	Total Sample			Matched		
Characteristic	NGT, *n* = 97 ^1^	Non-NGT, *n* = 243 ^1^	*p*-Value ^2^	NGT, *n* = 97 ^1^	Non-NGT, *n* = 97 ^1^	*p*-Value ^2^
duration of NGT therapy (days)	25 (19)	-		25 (19)	-	
length of stay (days)	39 (23)	31 (16)	0.002	39 (23)	36 (20)	0.3
discharge caloric intake	1501 (369)	1455 (318)	0.3	1501 (369)	1421 (321)	0.11
caloric intake increase	894 (483)	658 (418)	<0.001	894 (483)	795 (440)	0.14
discharge BMI	14.80 (1.63)	14.83 (1.73)	0.9	14.80 (1.63)	14.51 (1.69)	0.2
BMI increase	0.81 (0.74)	0.48 (0.72)	<0.001	0.81 (0.74)	0.55 (0.69)	0.016
WAI goal	24 (4)	24 (4)	0.6	24 (4)	24 (4)	0.3
WAI task	23 (5)	23 (5)	0.5	23 (5)	22 (5)	0.2
WAI bond	23 (4)	22 (5)	0.027	23 (4)	22 (5)	0.018
WAI total	71 (12)	69 (12)	0.2	71 (12)	67 (12)	0.082
EOT usefulness	76 (25)	74 (22)	0.6	76 (25)	73 (23)	0.4
EOT satisfaction	75 (23)	72 (23)	0.3	75 (23)	71 (23)	0.3
EOT subjective improvement	64 (24)	61 (25)	0.4	64 (24)	56 (25)	0.081
EOT meal assistance	61 (30)	57 (31)	0.3	61 (30)	53 (29)	0.12
EOT appropriateness length of stay	70 (26)	73 (26)	0.4	70 (26)	69 (27)	0.8
EDE-Q total reduction	0.58 (1.43)	0.66 (1.14)	0.7	0.58 (1.43)	0.81 (1.30)	0.3

^1^ Mean (SD); ^2^ Welch two-sample *t*-test; Abbreviations: BMI = Body Mass Index; EDE-Q = Eating Disorder Examination Questionnaire; NGT = Nasogastric Tube; WAI = Working Alliance Inventory; EOT = end of treatment.

**Table 3 nutrients-16-01664-t003:** Regression model with BMI increase as outcome.

Characteristic	β	95% CI ^1^	*p*-Value
admission BMI	−0.18	−0.24, −0.11	<0.001
NGT	—	—	
non-NGT	−0.32	−0.53, −0.10	0.005
discharge caloric intake	0.00	0.00, 0.00	<0.001
EDE-Q total reduction	0.03	−0.05, 0.11	0.5
length of stay (days)	0.00	−0.01, 0.01	0.8

^1^ CI = Confidence Interval; Abbreviations: BMI = Body Mass Index; EDE-Q = Eating Disorder Examination Questionnaire; NGT = Nasogastric Tube.

**Table 4 nutrients-16-01664-t004:** Regression models with end-of-treatment questionnaire evaluations as outcome.

Group	Characteristic	β	95% CI ^1^	*p*-Value
usefulness	BMI increase	6.5	0.70, 12	0.028
	NGT	—	—	
	non-NGT	1.3	−7.2, 9.7	0.8
	EDE-Q total reduction	3.8	0.29, 7.3	0.034
	length of stay (days)	0.06	−0.18, 0.29	0.6
	WAI total	0.97	0.62, 1.3	<0.001
	FMPS concern over mistakes	−0.13	−0.61, 0.35	0.6
	TEMPS-A depressive temperament	0.48	−0.70, 1.7	0.4
satisfaction	BMI increase	6.4	1.0, 12	0.020
	NGT			
	NGT	—	—	
	non-NGT	0.21	−7.5, 7.9	>0.9
	EDE-Q total reduction	4.0	0.80, 7.2	0.015
	length of stay (days)	0.05	−0.17, 0.26	0.7
	WAI total	0.93	0.61, 1.3	<0.001
	FMPS concern over mistakes	−0.33	−0.77, 0.12	0.15
	TEMPS-A depressive temperament	0.63	−0.45, 1.7	0.3
improvement	BMI increase	6.5	0.60, 12	0.031
	NGT			
	NGT	—	—	
	non-NGT	−3.8	−12, 4.7	0.4
	EDE-Q total reduction	4.4	0.90, 8.0	0.015
	length of stay (days)	−0.09	−0.32, 0.15	0.5
	WAI total	0.86	0.50, 1.2	<0.001
	FMPS concern over mistakes	0.13	−0.35, 0.62	0.6
	TEMPS-A depressive temperament	−1.1	−2.3, 0.10	0.071

^1^ CI = Confidence Interval; Abbreviations: BMI = Body Mass Index; EDE-Q = Eating Disorder Examination Questionnaire; WAI = Working Alliance Inventory; FMPS = Frost Multidimensional Perfectionism Scale; TEMPS-A = Temperament Evaluation of Memphis, Pisa, Paris, and San Diego Autoquestionnaire; NGT = Nasogastric Tube.

## Data Availability

The raw data supporting the conclusions of this article will be made available by the authors upon request.

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
