# Peer review of "Nasogastric Tube Feeding in Anorexia Nervosa: A Propensity Score-Matched Analysis on Clinical Efficacy and Treatment Satisfaction"

_nutrients, 2024, doi:10.3390/nu16111664_

Round 1

Reviewer 1 Report

Comments and Suggestions for Authors

This study is to compare a group of AN who underwent NGT nutrition during hospitalization with a group of propensity score matched AN who proceeded to entirely oral nutrition. 

Comment and questions

1. What is the exact indications for NGT nutrtion?

2. As your description, NGT nutrition was applied for sereve patients.

  Despite of the propensity matching, the difference might be present between two groups. How did you selecte the variables for matching. (I don't know the exact management or  baseline characteristic of AN patients)

admission BMI and caloric intake were not matched. 

How was a body weight at admission period? 

3. In NGT group, how many patients did supply by enteral and orally.

Author Response

This study is to compare a group of AN who underwent NGT nutrition during hospitalization with a group of propensity score matched AN who proceeded to entirely oral nutrition. 

Comment and questions

1.What is the exact indications for NGT nutrtion?

We now provide detailed description of indications for positioning of NGT: "According to the most recent American Psychiatric Association (APA) guidelines NGT feeding can be employed for acute nutritional rehabilitation in patients who fail to meet their prescribed caloric intake through meals alone and could also be influenced by factors such as age, other clinical characteristics, and the availability of specialized treatment programs like meal-based behavioral treatment (Crone et al., 2023) . Medical Emergencies in Eating Disorders (MEED) guidelines agree with using NGT only as a second resort and specify that it can become necessary when a patient cannot consume enough nutrients orally or cannot eat by mouth (Royal College of Psychiatrists, 2022). However, there are subtle differences in the approach to severely ill patients: the APA guidelines suggest that the decision to use NGT should not be solely based on medical instability or the severity of illness, whereas the MEED guidelines specifically list clinical/biochemical instability and life-threatening weight loss (Body Mass Index or BMI < 13) as criteria for NGT use."

2.As your description, NGT nutrition was applied for sereve patients.

Despite of the propensity matching, the difference might be present between two groups. How did you selecte the variables for matching. (I don't know the exact management or  baseline characteristic of AN patients)

As we now specify in the text, "These variables were selected according to literature evidence and clinical expertise". Although baseline differences in non-measured features could still be present after propensity score matching, we are confident that the non-NGT group has a comparable clinical and psychopathological severity in the considered key variables.

admission BMI and caloric intake were not matched. 

Due to pagination error in the table admission BMI and admission caloric intake were not reported on the same rows for non-matched and matched samples. We have now fixed the mistake and now is clear also from table 1 that samples were matched for these variables. We have fixed a similar error in table 2 as well. Furthermore, we have expanded the supplementary material with details on matching results.

How was a body weight at admission period? 

We have added mean weight (kg) to table 1 for all groups. The value is around 37 kg in every group.

  1. In NGT group, how many patients did supply by enteral and orally.

We now clarify in the text that patients in the NGT group received both oral and enteral nutrition during the inpatient stay. Additionally, we have specified the percentage of patients who began the combined approach from the start of their hospital stay: "NGT with daytime continuous nutritional supplementation was implemented along with the usual five oral meals per day, in an integrated manner. About 95% of patients with NGT commenced both oral and enteral nutrition at the start of their hospitalization, whereas the remaining 5% initially received only enteral feeding and transitioned to combined rehabilitation after a few days."

We now state in the limitation section that from the current dataset we cannot quantify the percentage of calories introduced via NGT and per os in the NGT group. However, this was beyond the scope of our study.

Reviewer 2 Report

Comments and Suggestions for Authors

In the manuscript, Martini et al. presented some interesting results that compared oral and NGT feeding efficiency in AN patients. Generally, the experiment design is sound, with extensive efforts on individual matching in sample groups. Noticeably, whilst the calorie intake at discharge was not significantly different in the matched groups (p=0.11), the outcome as measured by BMI demonstrated substantial improvements. The high efficiency in the NGT group is not surprising, considering that non-NGT AN patients usually have some degree of limiting factors (regurgitating, unable to finish the provided food quota, or failing to process the intake food, such as properly chewing down large food chunks that resulted in incomplete digestion, as observed clinically). Nevertheless, the manuscript provided a highly sophisticated analysis by regression that gave out easily comprehensible correlations between characteristics and their effective trend towards BMI increase, which is highly useful in analysing individual factors and their priority in future clinical applications.

Minor point: Figure S3-S6 should increase their Y-axes length for better data clarity. Currently, it is very difficult to interpret the Y-axis and their ticks.

Author Response

In the manuscript, Martini et al. presented some interesting results that compared oral and NGT feeding efficiency in AN patients. Generally, the experiment design is sound, with extensive efforts on individual matching in sample groups. Noticeably, whilst the calorie intake at discharge was not significantly different in the matched groups (p=0.11), the outcome as measured by BMI demonstrated substantial improvements. The high efficiency in the NGT group is not surprising, considering that non-NGT AN patients usually have some degree of limiting factors (regurgitating, unable to finish the provided food quota, or failing to process the intake food, such as properly chewing down large food chunks that resulted in incomplete digestion, as observed clinically). Nevertheless, the manuscript provided a highly sophisticated analysis by regression that gave out easily comprehensible correlations between characteristics and their effective trend towards BMI increase, which is highly useful in analysing individual factors and their priority in future clinical applications.

Thank you for your comments and insights.

Minor point: Figure S3-S6 should increase their Y-axes length for better data clarity. Currently, it is very difficult to interpret the Y-axis and their ticks

Thank you for pointing that out, we have improved the quality of the supplementary pictures that are now readable.

Reviewer 3 Report

Comments and Suggestions for Authors

This is an interesting, well-written paper on the role of EN as a method of nutritional support in ED.

 It has been shown that NGT nutrition allows for higher body weight gain (which depends on providing more calories in total), without a negative impact on the psychological parameters of patients.

I have a few comments:

Methods:

The enrollment started in 2014 whereas the italian version of „the 460 Eating Disorder Examination Questionnaire: Reliability and Validity of the Italian Version” applied and cited in the study was published in 2017.  Please explain this discrepancy.

The DAG (figure S1.) is not very readable. I suggest adding in the "methodology" section a short description-guide, explaining to readers the idea of DAG. Please provide a short summary of DAG you have created.

Table 2. Please add the discharge BMI value and average increase in daily caloric intake in the studied grups (NGT and non-NGT).

Author Response

This is an interesting, well-written paper on the role of EN as a method of nutritional support in ED.

 It has been shown that NGT nutrition allows for higher body weight gain (which depends on providing more calories in total), without a negative impact on the psychological parameters of patients.

I have a few comments:

Methods:

The enrollment started in 2014 whereas the italian version of „the 460 Eating Disorder Examination Questionnaire: Reliability and Validity of the Italian Version” applied and cited in the study was published in 2017.  Please explain this discrepancy.

The Italian version of the EDE-Q was available before the validation study published in 2017, therefore there was no differences in the instrument that participant filled out before and after that date.

The DAG (figure S1.) is not very readable. I suggest adding in the "methodology" section a short description-guide, explaining to readers the idea of DAG. Please provide a short summary of DAG you have created.

We have now added description of causal paths and definition of DAG to the method section: "DAGs are visual representations of the variables that researchers assume to be responsible for the data-generation process of interest (Tennant et al., 2021). In such graphs, variables are represented as nodes, and the causal relationships between them as arrows. Such representation allow to define the causal path that moves from the exposure to the outcome, as well as the so called biasing paths due to confounders. Such paths could arise, for instance, when both the exposure and the outcome are caused by a third variable. Taking into account these relationship in the analysis is needed in order avoid biased estimates of the effect of exposure on the outcome.In s imple terms, DAGS provide a way to 1) transparently state assumed causal relations, and 2) identify confounders in the causal relationship of interest. As a practical implication, by inspecting a DAG (or using programs dedicated to draw and anayze DAGs such as dagitty; Tennant et al., 2021), researchers can easily define the sufficient adjustment set for their analysis (i.e., the variables to condition on in order to "close non-causal paths")."

The legend of supplementary figures explains how to interpret the DAGs: "The exposure is depicted as a green circle with a black triangle in the middle. The outcome is represented by a blue circle with a black rectangle in the middle.  Red circles are ancestor of both exposure and outcome, whereas blue circles are ancestor of outcome. White circles are the adjusted variable, and grey other variables not influencing the causal path of interest. The green arrow identifies the causal path. Biasing paths would appear as purple arrows. In this case, multiple biasing path would be evident if adjusted variables would not be defined as such (i.e., not conditioned on in the analysis)."

Table 2. Please add the discharge BMI value and average increase in daily caloric intake in the studied grups (NGT and non-NGT).

We have added mean discharge BMI and mean caloric intake increase in table 2 for all groups.

Round 2

Reviewer 1 Report

Comments and Suggestions for Authors

Authors documented :. According to international guidelines [8], oral 150 nutritional rehabilitation was the preferred refeeding method, however, enteral therapy 151 was considered for severe cases, typically when significant nutritional intake challenges 152 were encountered early in hospitalization"

-> Were there any indications or  decision making process of NGT insertion for the AN patients? If you can add the indications of your institution or algorithm for NGT insertion process. I think this can be helpful to readers to adopt the NGT insertion.

However, if you don't have any indicaitons, or NGT insertion was made by the physician's decision, add in the method section and describe in the limitation for the vague indication

Author Response

Authors documented :. According to international guidelines [8], oral 150 nutritional rehabilitation was the preferred refeeding method, however, enteral therapy 151 was considered for severe cases, typically when significant nutritional intake challenges 152 were encountered early in hospitalization"

-> Were there any indications or  decision making process of NGT insertion for the AN patients? If you can add the indications of your institution or algorithm for NGT insertion process. I think this can be helpful to readers to adopt the NGT insertion.

However, if you don't have any indicaitons, or NGT insertion was made by the physician's decision, add in the method section and describe in the limitation for the vague indication

Following your suggestion, we have added in the Method section: "The decision of prescribing nutritional therapy via NGT was discussed within the multidisciplinary team and with the patient, and NGT therapy was administered by clinicians with years of expertise in inpatient treatment of AN."
Furthermore, as we specify in the causal assumptions section, the proximal reasons for NGT positioning are not documented in the database, therefore we focused our analysis on the direct effects of NGT therapy on the outcomes. 
In the limitations we now specify that "The decision of NGT positioning was made by clinicians and did not follow standardized decisional algorithms; however, we focused our analysis on the effect of enteral therapy on the outcomes, while taking into account potential baseline and end-of-treatment confounders."